# The Combination Effect of the Red Blood Cell Distribution Width and Prognostic Nutrition Index on the Prognosis in Patients Undergoing PCI

**DOI:** 10.3390/nu16183176

**Published:** 2024-09-19

**Authors:** Likun Huo, Wenjuan Zhao, Xiang Ji, Kangyin Chen, Tong Liu

**Affiliations:** 1Department of Emergency, Tianjin Huanhu Hospital, Tianjin 300222, China; huolikungege@126.com (L.H.);; 2Tianjin Key Laboratory of Ions and Molecular Function of Cardiovascular Diseases, Department of Cardiology, Tianjin Institute of Cardiology, The Second Hospital of Tianjin Medical University, Tianjin 300211, China

**Keywords:** coronary heart disease, prognostic nutrition index, red blood cell distribution width, percutaneous coronary intervention, prognosis

## Abstract

Background: Inflammation and malnutrition are related to adverse clinical outcomes in patients with coronary artery disease (CAD). However, it is unclear whether there is a relationship between the PNI (prognostic nutritional index) and RDW (red blood cell distribution width) regarding the impact on the prognosis in patients with CAD undergoing percutaneous coronary intervention (PCI). Methods: A total of 5605 consecutive CAD patients undergoing PCI were selected retrospectively. The patients were stratified into four groups according to the PNI [high PNI (H-PNI) and low PNI (L-PNI)] and RDW [high RDW (H-RDW) and low RDW (L-RDW)]. The cutoff values of RDW and PNI were calculated using receiver-operating characteristic curve analysis. The primary endpoint was 1-year all-cause mortality (ACM). The secondary endpoint was major adverse cardiac cerebrovascular events (MACCEs), the composite of cardiac death (CD), the recurrence of MI, target lesion revascularization (TLR), and stroke. A Cox proportional hazards model was used to evaluate the association between the PNI, RDW, and clinical endpoints. Results: During 1-year follow-up, 235 (4.19%) patients died. In multivariate regression analysis, the L-PNI/H-RDW group was found to have the highest risk of 1-year ACM [hazard ratio (HR) = 8.85, 95% confidence interval (CI): 5.96–13.15, *p* = 0.020] with the H-PNI/L-RDW group as a reference, followed by the L-PNI/L-RDW (HR = 3.96, 95% CI: 2.60–6.00, *p* < 0.001) and H-RDW/H-PNI groups (HR = 3.00, 95% CI: 1.99–4.50, *p* < 0.001). Nomograms were developed to predict the probability of 1-year ACM and MACCEs. Conclusions: CAD patients with L-PNI and H-RDW experienced the worst prognosis. The combination of PNI and RDW was a strong predictor of 1-year ACM. The coexistence of PNI and RDW appears to have a synergistic effect, providing further information for the risk stratification of CAD patients.

## 1. Introduction

Coronary artery disease (CAD) is a leading public health problem that has a huge burden on society. Over the past couple of years, percutaneous coronary intervention (PCI) has become one of the most widely used treatment strategies for patients. Although the PCI technique improves adverse prognosis, several individuals are still at risk for adverse cardiovascular events. Early and easy-to-use risk stratification for prognosis is necessary for the individualized management of patients post-PCI. Malnutrition is usually caused by imbalanced and inadequate nutrient intake and contributes to the lack of calories or proteins, which is related to many diseases. CAD may cause complex physical and psychological changes and affect nutritional status. However, clinicians often pay little attention to assessing nutritional status. In fact, malnutrition is not unusual in patients with CAD.

Previous studies have shown that malnutrition is strongly associated with increased mortality and adverse cardiovascular events [1,2]. The prognostic nutritional index (PNI) is a simple and accessible index that can allow us to assess nutritional status quantitatively. The PNI is calculated as serum albumin (g/L) + 5 × total lymphocyte count. It is a comprehensive index that reflects not only protein stores but also the immunological status. In recent years, many studies have shown that the PNI is an adverse predictor of cardiovascular disease [3,4]. Inflammation is a key mediator in the impact of malnutrition on the prognosis of CAD [5] and plays an important role in the development and progression of atherosclerosis. Several studies have shown that the RDW is associated with a higher risk of mortality and adverse cardiac events in patients with CAD undergoing PCI [6].

Both malnutrition and RDW have been shown to be associated with worse clinical outcomes in patients with CAD. However, most previous studies focused solely on inflammation markers or malnutrition. Accordingly, this study aimed to explore the clinical value of the combination of the RDW and PNI on admission in predicting clinical outcomes after PCI with respect to improving risk stratification.

## 2. Materials and Methods

### 2.1. Study Patients and Design

A retrospective cohort study was conducted. A total of 5605 patients with CAD who underwent percutaneous coronary intervention (PCI) admitted to the Second Hospital of Tianjin Medical University from 1 January 2019 to 30 June 2022 were selected.

The inclusion criteria were as follows: (1) patients aged ≥18 years; (2) patients who received PCI. The exclusion criteria were as follows: (1) patients with known malignancy or active inflammatory disease; (2) serious hepatic dysfunction or renal insufficiency; (3) the concurrence of autoimmune diseases. The population was divided into the following 4 groups: high-PNI/low-RDW(H-PNI/L-RDW), high-PNI/high-RDW (H-PNI/H-RDW), low-PNI/low-RDW (L-PNI/L-RDW), and low-PNI/high-RDW (L-PNI/H-RDW). The optimal cutoff values of PNI and RDW for 1-year ACM were obtained from receiver-operating characteristic curve analysis.

This study protocol was approved by the ethics committee of the Second Hospital of Tianjin Medical University and complied with the Declaration of Helsinki. Due to the retrospective design of the study, informed consent from eligible patients was waived by the ethics committee.

### 2.2. Demographic and Clinical Data Collection

Data on demographic and clinical characteristics were collected from medical records; these included age; sex; comorbidities such as hypertension, diabetes, dyslipidemia, atrial fibrillation (AF), peripheral artery disease (PAD), and chronic kidney disease (CKD); current smoking status; and previous history, such as myocardial infarction (MI), previous stroke, and previous PCI. The laboratory data included plasma and biochemical parameters, such as the levels of white blood cells (WBCs), hemoglobin (Hb), absolute lymphocyte count (Lym), red blood cell distribution width (RDW), serum albumin (Alb), triglycerides (TG), total cholesterol (TC), low-density lipoprotein cholesterol (LDL-C), creatinine (Cr), and uric acid (UA).

### 2.3. Endpoints and Follow-Up

The primary endpoint was 1-year all-cause mortality (ACM). The secondary endpoint was major adverse cardiac cerebrovascular events (MACCEs), the composite of cardiac death (CD), recurrence of MI, target lesion revascularization (TLR), and stroke.

## 3. Statistical Analysis

R software version 4.2.1 (R Foundation for Statistical Computing, Vienna, Austria) was used for statistical analysis. The Kolmogorov–Smirnov test was used to evaluate the normal distribution of the data. Continuous data were presented as the mean ± standard deviation (SD) or a median with interquartile range (IQR), and categorical data were expressed as *n* (%). Mann–Whitney U tests, Kruskal–Wallis tests, or χ^2^ tests were used to compare baseline characteristics, as appropriate. The optimal cutoff values of PNI and RDW for 1-year ACM were obtained from receiver operating characteristic curve analysis. The patients were divided into four groups, which were categorized on the baseline of PNI and RDW. Kaplan–Meier analysis and a log-rank test were used to estimate the cumulative incidence of clinical outcomes. Univariate and multivariate Cox regression analyses were performed to identify the association of PNI and RDW with clinical outcomes. Variables with *p* < 0.1 in the univariate analysis and other clinically relevant variables based on the previous studies were analyzed in the multivariate Cox regression. Subgroup analysis for 1 year ACM was conducted in diverse subgroups (age ≥ 65 years/age < 65 years, male/female, Hb ≥ 90 g/L/Hb < 90 g/L, MI/Non-MI). Nomograms were performed to predict the probability of 1-year ACM and MACCEs based on the results of the multivariable COX regression. *p* values < 0.05 were considered statistically significant.

## 4. Results

### 4.1. Baseline Characteristics

A total of 5605 patients with CAD who underwent PCI were included in our study (Figure 1). As shown in Table 1, 63.4% of the participants were male, and the median age was 66 years [IQR 60, 73]. The median values of PNI and RDW were 49.4 (45.8–53.0) and 13.0 (12.6–13.8) for all participants, respectively.

When grouping by 1-year vital status, patients in the non-survival group were older; had more comorbidities (hypertension, peripheral artery disease, AF, and CKD); were more likely to have a previous history of MI and stroke; and had lower Hb, lymphocyte counts, and Alb. The PNI was significantly lower and the RDW was higher in non-survival group patients. There were more patients with myocardial infarction than in the survival group.

The optimal cutoff values of the PNI and RDW for predicting 1-year all-cause mortality were 44.2 and 13.6, respectively. Table 2 shows the baseline characteristics among the four groups stratified by PNI and RDW level. The patients in the L-PNI/H-RDW group had more comorbidities (hypertension, DM, AF, peripheral artery disease, and CKD) than other groups and were more likely to have a previous history of stroke. They were also likely to have lower Hb, lymphocyte counts, and Alb.

### 4.2. Relationship between PNI Combined with RDW and Clinical Outcomes

Death and MACEE occurred in 4.19% (235/5605) and 10.0% (560/5605) of patients at 1 year, respectively. Kaplan–Meier survival curves were plotted based on the cutoff value of the PNI and RDW. The incidence of 1-year ACM was higher in the low-PNI group compared to the high-PNI group. Meanwhile, patients with an isolated high RDW had a higher incidence of 1-year ACM than the low-RDW group (*p* < 0.001, Figure 2A,B). Patients in the L-PNI/H-RDW group had the highest incidence of 1-year ACM among the four groups (1.3% vs. 4.5% vs. 8.8% vs. 20.8%, *p* < 0.001, Figure 2C). Similarly, compared to patients in the isolated high-PNI or low-RDW groups, the cumulative rate of MACCEs was higher in patients with a low PNI or high RDW (*p* < 0.001, Figure 3A,B). Patients in the L-PNI/H-RDW group had the highest incidence of MACCEs at 1 year among the four groups (6.4% vs. 12.5% vs. 15.0% vs. 25.4%, *p* < 0.001, Figure 3C).

Univariate and multivariate analyses for 1-year all-cause mortality are presented in Table 3. After adjustment, the L-PNI/H-RDW group had a higher risk of 1-year all-cause mortality compared to the H-PNI/L-RDW group (HR: 8.85, 95% CI: 5.96–13.15, *p* = 0.020), followed by the L-PNI/L-RDW group (HR: 3.96, 95% CI: 2.60–6.00, *p* < 0.001) and the H-PNI /H-RDW group (HR: 3.00, 95% CI: 1.99–4.50, *p* < 0.001).

For 1-year MACCEs, univariate and multivariate Cox analyses also showed that the L-PNI/H-RDW group had a significantly higher risk, as indicated in Table 4.

### 4.3. Subgroup Analysis

The 1-year ACM stratified by the combination of the PNI and RDW level was assessed in diverse subgroups (age ≥ 65 years/age < 65 years, male/female, Hb ≥ 90 g/L/Hb < 90 g/L, MI/Non-MI) (Figure 4).

Subgroup analyses by sex and presentation status showed consistent results, with the primary analysis that the L-PNI/H-RDW group had the highest mortality risk. There were no significant interactions between subgroups (all *p* for interaction > 0.05), except for age. Similarly, 1-year MACCEs stratified by the combination of PNI and RDW levels were assessed in diverse subgroups (Figure 5). In male patients aged ≥ 65 years, Hb ≥ 90 g/L, and Non-MI, the L-PNI/H-RDW had a higher risk of MACCEs. However, there were no significant interactions in the subgroup analysis.

### 4.4. A Novel Nomogram for 1-Year Adverse Prognosis via PCI

Nomograms were developed to predict the probability of one-year ACM and MACEEs, which include the PNI, RDW, and other significant predictors mentioned earlier (Figure 6). The nomogram enabled the direct reading of the probability for predicting adverse clinical outcomes after summing the points for each predictor.

## 5. Discussion

In the present study, we evaluated the impact of the combination of the RDW and PNI on adverse clinical outcomes after PCI. The major findings were as follows: (1) Patients with a high RDW or low PNI experienced a poor prognosis when they had CAD when undergoing PCI. (2) Patients with L-PNI/H-RDW experienced the highest risk of ACM and MACCEs at 1 year, and the coexistence of PNI and RDW is an independent predictor. (3) The PNI and RDW appear to have a synergistic effect.

Malnutrition is common in patients with CAD and is strongly associated with increased mortality and cardiovascular events [7]. A number of nutritional indices have been used to assess nutritional status, such as the Controlling Nutritional Status (CONUT) score, the Geriatric Nutritional Risk Index (GNRI), and the PNI.

The calculation of the CONUT score requires the serum cholesterol level in addition to the albumin level and lymphocyte count, which may present limitations in patients with CAD [8]. GNRIs are calculated by combining albumin and body mass index (BMI) values and may underestimate malnutrition in individuals with normal or excessive BMI [9]. Therefore, the PNI is a simple and effective tool for assessing malnutrition in patients with CAD.

The prognostic nutrition index (PNI), proposed by Mullen and his colleagues, was used to assess the prognosis of patients undergoing gastrointestinal surgery [10]. The PNI was calculated from serum albumin levels and absolute lymphocyte counts, which was a comprehensive index that reflected not only protein stores but also the immunological status. Several researchers have proven that the PNI is associated with a poor prognosis in patients with a variety of diseases, including cancer, lymphoma, infectious diseases, and postoperative complications [11]. As an important component of PNI, serum albumin is an important antioxidant outside the cell. When serum albumin remains at normal concentrations, it plays an important role in inhibiting platelet activation and aggregation and vascular endothelial apoptosis [12]. Hypoalbuminemia, which is related to weakened antioxidant and antithrombotic capacities of albumin, mainly raises cardiovascular risk [13]. Furthermore, a decrease in serum albumin indicates potential inflammation, which causes the progression of atherosclerosis [14]. Previous studies reported that low serum albumin levels are a strong predictor of all-cause mortality in patients with ACS, even after adjusting for common confounders [15]. However, there is a lack of clinical trials demonstrating that the correction of serum albumin levels via intravenous infusion reduces the risk of excessive mortality in patients with ACS [16]. Low lymphocyte counts reflect poorly regulated immune responses and are also associated with poor outcomes in patients with CAD [17]. The PNI, which combines albumin levels and lymphocyte counts, reflects the immune-nutritional status of the human body. A growing body of research has demonstrated associations between clinical outcomes and malnutrition in patients with heart failure, ST-segment-elevated myocardial infarction (STEMI), non-ST-segment elevation myocardial infarction (NSTEMI), and stable coronary artery disease (CAD).

A recent study including 437 patients with NSTEMI suggested that the PNI is significantly associated with in-hospital mortality and the GRACE risk score [18]. In a retrospective study, a lower PNI was associated with an increased risk of all-cause mortality in AMI patients admitted to the ICU after adjusting for potential risk factors. The PNI was a convincing predictor of 6-month and 1-year all-cause mortality [19]. A study conducted on 309 patients with STEMI undergoing PCI showed that the PNI was a significant independent predictor of mortality. In this study, compared with the high-PNI group (PNI > 45), the low-PNI group (PNI ≤ 45) showed higher mortality (21.7% vs. 6.9%, *p* < 0.001) at long-term follow-up [20]. In a retrospective study including patients who underwent PCI with CAD PNI was significantly associated with long-term cardiovascular outcomes [21]. In this study, patients with a low PNI have a higher frequency of not only all-cause mortality but also cardiac mortality, but with a small sample size.

Similarly, a prospective study from China showed that the PNI was a new biomarker that could be used to predict the long-term outcome of CAD patients after PCI [22]. Our study found that the PNI level was lower in the non-survival and MACCE groups, which is in line with previous studies.

Research has shown that malnutrition is strongly associated with inflammation and atherosclerosis in patients with end-stage renal disease, and the interaction of malnutrition and inflammation may also exacerbate poor outcomes in patients with CAD.

The red blood cell distribution width (RDW) is easy to obtain and represents the coefficient of variation of the red blood cell volume distribution width. Many mechanisms can affect RDW levels, such as inflammatory stress, adrenergic activation, nutritional deficiency, and iron homeostasis disorder. Previous studies showed that an increased RDW is a powerful independent predictor of cardiovascular events in heart disease patients [23,24].

Blood indicators are simple and easy to obtain, but a single factor cannot easily and accurately predict the prognosis. Combined markers of inflammation and malnutrition, such as the RDW and PNI, are warranted.

Our research indicates that the combination of two indices is superior to a single index in predicting outcomes for CAD patients undergoing PCI. This biomarker provides evidence for risk stratification in patients with CAD. There were some limitations to our study. This is a single-center and retrospective study, which could have caused selection bias. Secondly, we only measured the admission data of the RDW and PNI; the influence of dynamic changes on mortality was unclear. The effect of the PNI and RDW in prognostic prediction needs to be further identified by a large sample and multicenter prospective studies.

## 6. Conclusions

CAD patients with H-RDW and L-PNI experienced the worst prognosis. H-RDW/L-PNI has better discrimination and prognostic abilities than the individual indices. The coexistence of RDW and PNI appears to have a synergistic effect, providing further information for stratifying the risk of CAD patients.

## Figures and Tables

**Figure 1 nutrients-16-03176-f001:**
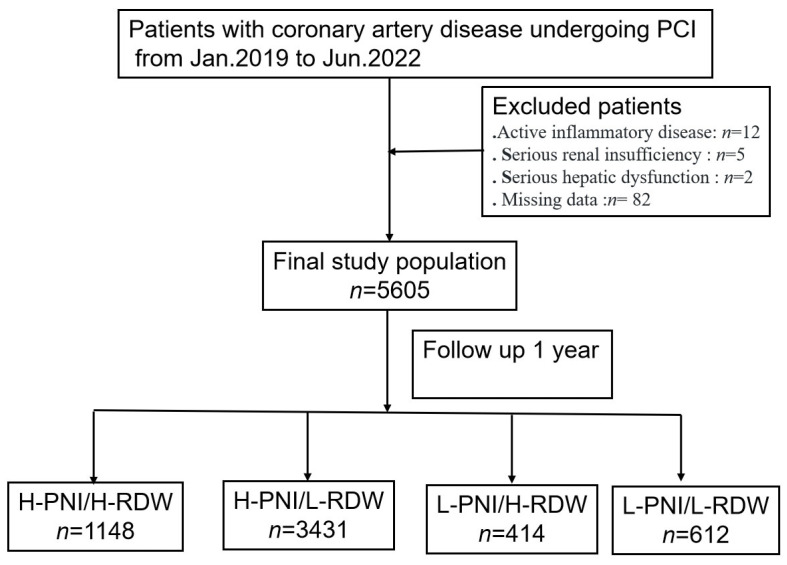
The study flowchart.

**Figure 2 nutrients-16-03176-f002:**
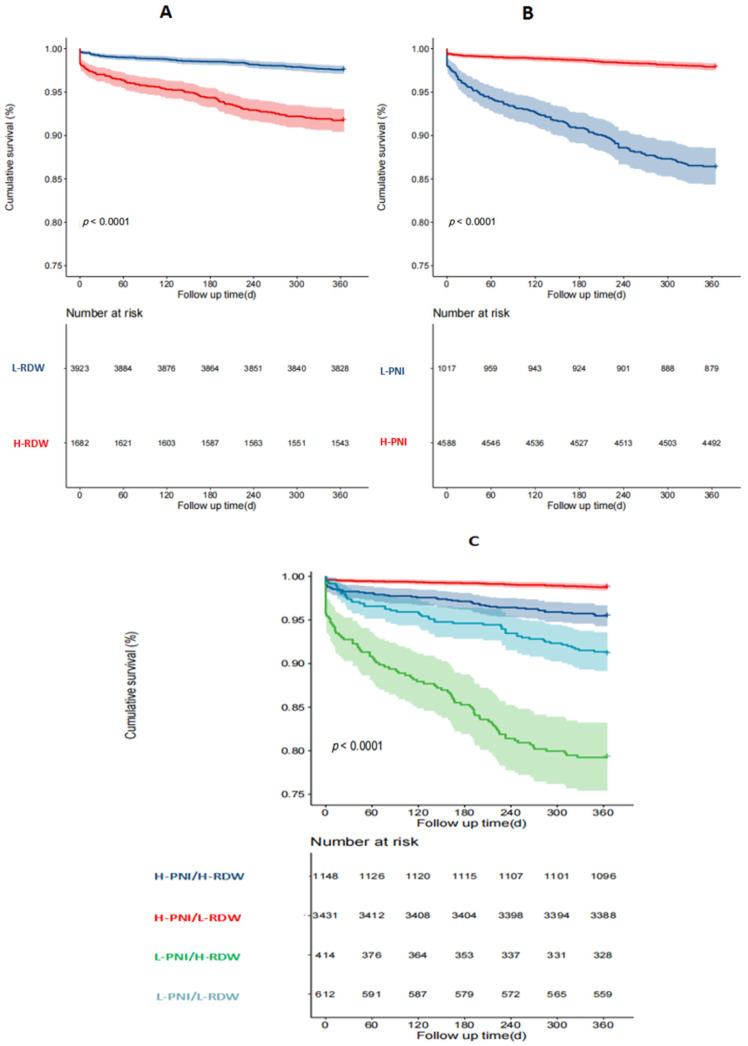
Kaplan–Meier analysis for all-cause mortality. The survival curves are stratified by the RDW level (**A**), PNI level (**B**), and both (**C**).

**Figure 3 nutrients-16-03176-f003:**
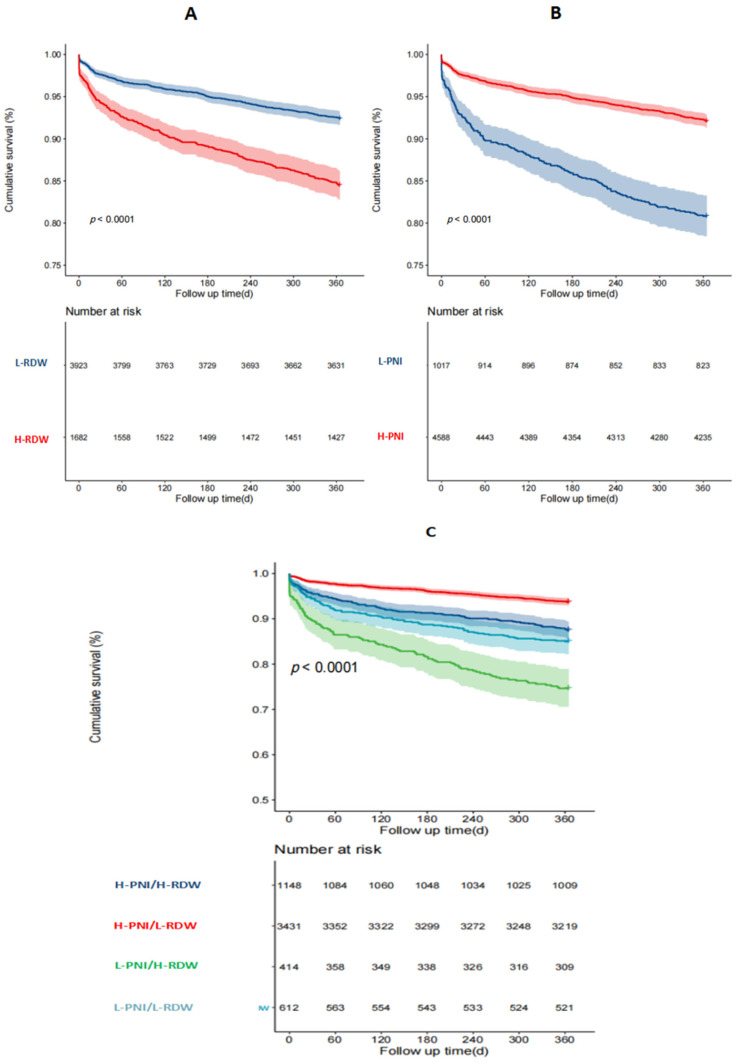
Kaplan–Meier analysis for 1-year MACCEs. The survival curves are stratified by the RDW level (**A**), PNI level (**B**), and both (**C**).

**Figure 4 nutrients-16-03176-f004:**
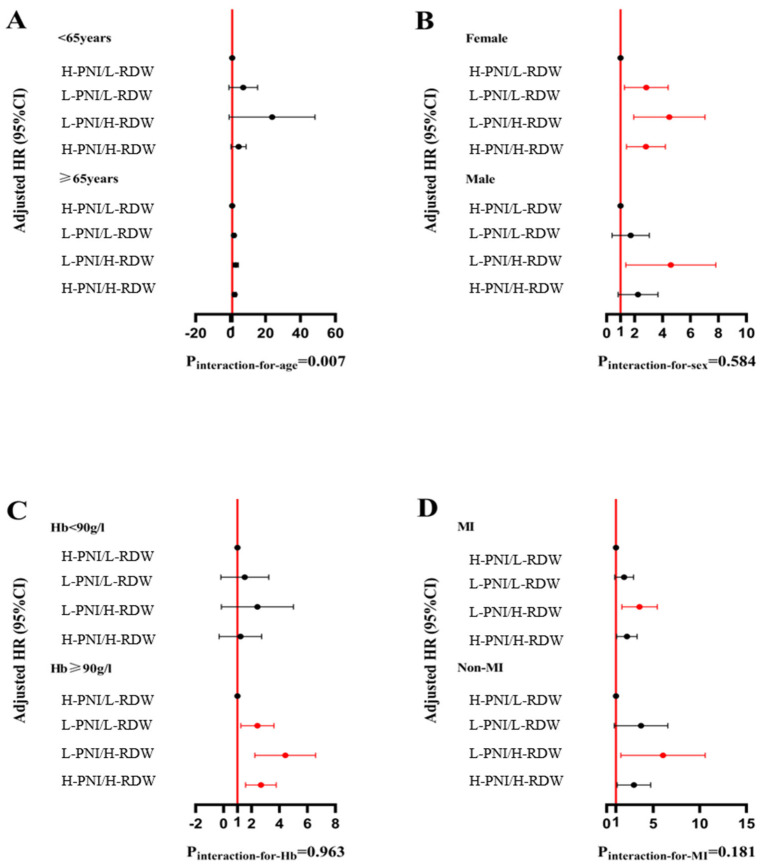
The association of the RDW and PNI level with all-cause mortality. In the Forrest plots for subgroups, red indicates statistical significance, while black indicates statistical non-significance. Subgroups were defined by age (**A**), sex (**B**), Hb category (**C**), and admission presentation (**D**).

**Figure 5 nutrients-16-03176-f005:**
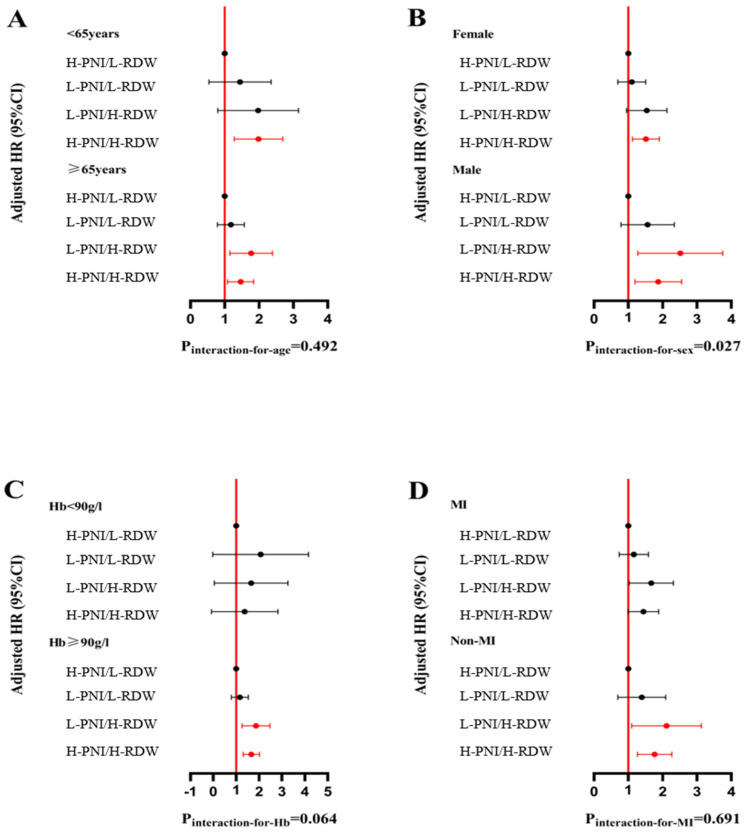
The association of the RDW and PNI level with 1-year MACCEs. In the Forrest plots for subgroups, red indicates statistical significance, while black indicates statistical non-significance. Subgroups were defined by age (**A**), sex (**B**), Hb category (**C**), and admission presentation (**D**).

**Figure 6 nutrients-16-03176-f006:**
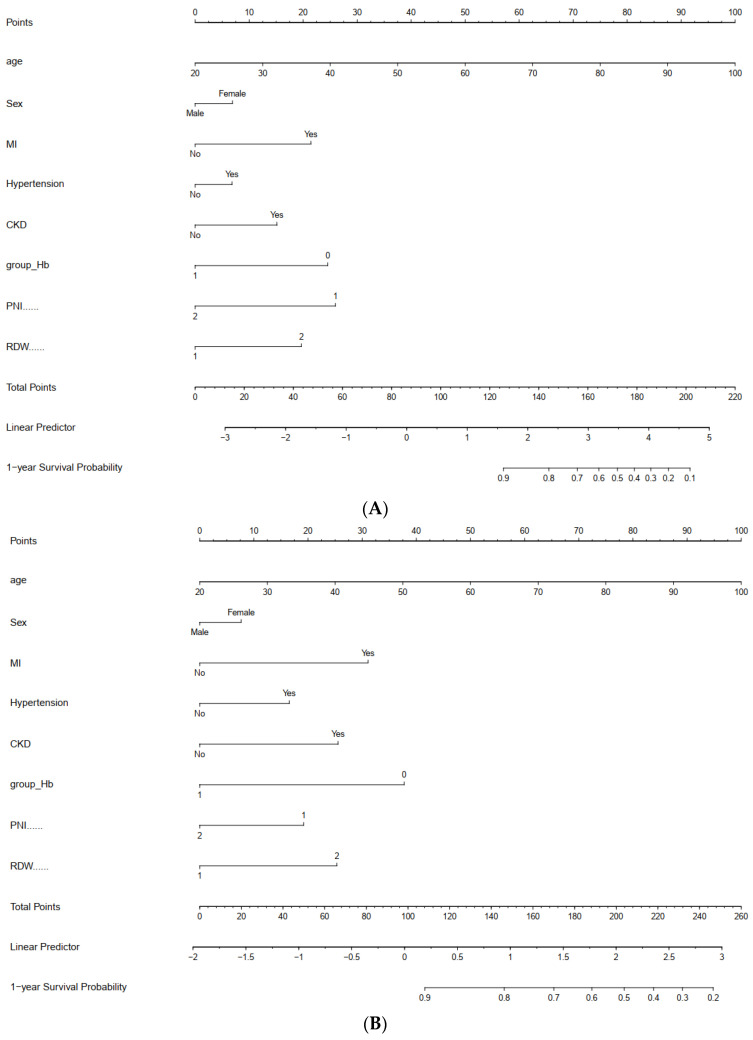
Nomograms for 1-year ACM (**A**) and MACCEs (**B**) undergoing PCI.

**Table 1 nutrients-16-03176-t001:** The baseline characteristics stratified by vitals.

Variable	All Participants (*n* = 5905)	Survival (*n* = 5370)	Non-Survival (*n* = 235)	*p*
Age, years	66.0 [60.0, 73.0]	66.0 [60.0, 72.0]	73.0 [66.0, 80.0]	<0.001
Male, *n* (%)	3554 (63.4%)	3397 (63.3%)	157 (66.8%)	0.300
MI, *n* (%)	1933 (34.5%)	1780 (33.1%)	153 (65.1%)	<0.001
STEMI, *n* (%)	1049 (18.7%)	971 (18.1%)	78 (33.2%)	<0.001
Non-STEMI, *n* (%)	884 (15.8%)	809 (15.1%)	75 (31.9%)	<0.001
Pre-PCI, *n* (%)	1254 (22.4%)	1210 (22.5%)	44 (18.7%)	0.197
Pre-MI, *n* (%)	750 (13.4%)	708 (13.2%)	42 (17.9%)	0.049
Pre-AF, *n* (%)	221 (3.94%)	202 (3.76%)	19 (8.09%)	0.002
Hypertension, *n* (%)	4089 (73.0%)	3897 (72.6%)	192 (81.7%)	0.003
DM, *n* (%)	2029 (36.2%)	1930 (35.9%)	99 (42.1%)	0.063
Dyslipidemia, *n* (%)	654 (11.7%)	630 (11.7%)	24 (10.2%)	0.544
Pre-IS, *n* (%)	829 (14.8%)	768 (14.3%)	61 (26.0%)	<0.001
PAD, *n* (%)	87 (1.55%)	81 (1.51%)	6 (2.55%)	0.180
CKD, *n* (%)	247 (4.41%)	201 (3.74%)	46 (19.6%)	<0.001
Current smoker, *n* (%)	1499 (26.7%)	1435 (26.7%)	64 (27.2%)	0.922
Hb, g/L	136 [124, 148]	137 [125, 149]	119 [97.0, 135]	<0.001
Lmy, 10^9^/L	1.64 [1.23, 2.10]	1.65 [1.25, 2.11]	1.25 [0.88, 1.77]	<0.001
RDW, %	13.0 [12.6, 13.8]	13.0 [12.6, 13.7]	14.0 [13.1, 16.3]	<0.001
UA, μmol/L	344 [283, 410]	343 [283, 408]	373 [300, 486]	<0.001
Cr, μmol/L	74.7 [61.9, 93.9]	74.1 [61.6, 91.9]	105 [71.0, 215]	<0.001
WBC, 10^9^/L	7.25 [5.96, 8.96]	7.22 [5.95, 8.92]	8.07 [6.11, 10.6]	<0.001
TC, mmol/L	4.55 [3.79, 5.26]	4.56 [3.80, 5.28]	4.32 [3.48, 4.98]	<0.001
TG, mmol/L	1.46 [1.05, 1.97]	1.46 [1.06, 1.98]	1.30 [0.93, 1.71]	<0.001
LDL, mmol/L	2.89 [2.26, 3.46]	2.90 [2.26, 3.47]	2.77 [2.10, 3.20]	0.014
Alb, g/L	40.6 [38.0, 43.3]	40.6 [38.3, 43.4]	36.2 [33.2, 39.6]	<0.001
PNI	49.4 [45.8, 53.0]	49.4 [46.0, 53.2]	42.6 [38.7, 48.1]	<0.001

The values are presented as a number (%) or median [interquartile range]. Abbreviations: Alb, serum albumin; Cr, creatine; CKD, chronic kidney disease; DM, diabetes; Hb, hemoglobin; Lmy, lymphocyte; LDL, low-density lipoprotein; MI, myocardial infarction; Non-STEMI, non-ST-segment elevation myocardial infarction; PNI, prognostic nutritional index; PAD, peripheral artery disease; Pre-IS, previous ischemic stroke; Pre-MI, previous myocardial infarction; Pre-AF, previous atrial fibrillation; Pre-PCI, previous percutaneous coronary intervention; RDW, red blood cell distribution width; STEMI, ST-elevated myocardial infarction; TC, total cholesterol; TG, triglycerides; UA, uric acid; WBC, white blood cell.

**Table 2 nutrients-16-03176-t002:** The baseline characteristics stratified by prognostic nutrition index and red blood cell distribution width.

Variable	All *n* = 5605	H-PNI/H-RDW *n* = 1148	H-PNI/L-RDW *n* = 3431	L-PNI/L-RDW *n* = 612	L-PNI/H-RDW *n* = 414	*p*
Age, years	66.0 [60.0, 73.0]	65.0 [58.0, 71.0]	66.0 [60.0; 72.0]	70.0 [64.0, 78.0]	66.0 [65.0, 78.0]	<0.001
Male, *n* (%)	3554 (63.4%)	730 (63.6)	2070 (62.0%)	424 (69.3)	269 (65.0)	0.012
MI, *n* (%)	1933 (34.5%)	446 (38.9%)	873 (26.1%)	332 (57.3%)	239 (58.2%)	<0.001
STEMI, *n* (%)	1049 (18.7%)	255 (22.2%)	488 (14.6%)	168 (29.0%)	114 (27.7%)	<0.001
Non-STEMI, *n* (%)	884 (15.8%)	191 (16.6%)	385 (11.5%)	164 (28.3%)	125 (30.4%)	<0.001
Pre-PCI, *n* (%)	1254 (22.4%)	234 (20.4%)	758 (22.7%)	128 (22.1%)	86 (20.9%)	<0.001
Pre-MI, *n* (%)	750 (13.4%)	149 (13.0%)	414 (12.4%)	81 (14.0%)	82 (20.0%)	<0.001
Pre-AF, *n* (%)	221 (3.94%)	56 (4.88%)	90 (2.70%)	35 (6.04%)	34 (8.27%)	<0.001
Hypertension, *n* (%)	4089 (73.0%)	827 (72.0%)	2408 (72.1%)	421 (72.7%)	328 (79.8%)	0.002
DM, *n* (%)	2029 (36.2%)	407 (35.5%)	1177 (35.3%)	224 (38.7%)	179 (43.6%)	0.010
Dyslipidemia, *n* (%)	654 (11.7%)	131 (11.4%)	390 (11.7%)	72 (12.4%)	49 (11.9%)	0.897
Pre-IS, *n* (%)	829 (14.8%)	176 (15.3%)	442 (13.2%)	101 (17.4%)	91 (22.1%)	<0.001
PAD, *n* (%)	87 (1.55%)	29 (2.53%)	38 (1.14%)	9 (1.55%)	11 (2.68%)	0.003
CKD, *n* (%)	247 (4.41%)	51 (4.44%)	47 (1.41%)	54 (9.33%)	90 (21.9%)	<0.001
Current smoker, *n* (%)	1499 (26.7%)	343 (29.9%)	854 (25.6%)	148 (25.6%)	116 (28.2%)	0.049
Hb, g/L	136 [124, 148]	135 [121, 147]	140 [130, 151]	127 [112, 140]	114 [95.0, 130]	<0.001
Lym, 10^9^/L	1.64 [1.23, 2.10]	1.77 [1.44, 2.30]	1.76 [1.39, 2.19]	1.05 [0.79, 1.31]	1.04 [0.78, 1.35]	<0.001
RDW, %	13.0 [12.6, 13.8]	16.4 [14.1, 41.1	12.8 [12.5, 13.1]	12.8 [12.5, 13.1]	15.2 [14.2, 41.7]	<0.001
UA, μmol/L	344 [283, 410]	354 [288, 417]	337 [281, 399]	351 [283, 427]	372 [299, 458]	<0.001
Cr, μmol/L	74.7 [61.9, 93.9]	76.0 [62.2, 97.8]	71.3 [60.7, 85.2]	85.5 [66.2, 128]	107 [74.3, 230]	<0.001
WBC, 10^9^/L	7.25 [5.96, 8.96]	7.76 [6.42, 9.43]	7.05 [5.88, 8.63]	7.12 [5.78, 9.54]	7.40 [5.74, 9.36]	<0.001
TC, mmol/L	4.55 [3.79, 5.26]	4.58 [3.88, 5.27]	4.58 [3.84, 5.34]	4.20 [3.56, 4.89]	4.24 [3.58, 4.90]	<0.001
TG, mmol/L	1.46 [1.05, 1.97]	1.54 [1.11, 2.01]	1.50 [1.10, 2.06]	1.20 [0.88, 1.67]	1.23 [0.91, 1.71]	<0.001
LDL, mmol/L	2.89 [2.26, 3.46]	2.91 [2.31, 3.49]	2.91 [2.29, 3.52]	2.64 [2.08, 3.26]	2.67 [2.08, 3.13]	<0.001
Alb, g/L	40.6 [38.0, 43.3]	40.6 [39.0, 42.8]	41.8 [39.7, 44.4]	35.8 [33.9, 37.7]	34.9 [32.7, 37.0]	<0.001
PNI	49.4 [45.8, 53.0]	49.4 [47.5, 53.1]	50.8 [48.1, 54.2]	41.7 [39.5, 43.0]	40.9 [38.0, 42.6]	<0.001

The values are presented as a number (%) or median [interquartile range]. Abbreviations: Alb, serum albumin; Cr, creatine; CKD, chronic kidney disease; DM, diabetes; Hb, hemoglobin; Lym, lymphocyte; LDL, low-density lipoprotein; MI, myocardial infarction; Non-STEMI, non-ST-segment elevation myocardial infarction; PNI, prognostic nutritional index; PAD, peripheral artery disease; Pre-IS, previous ischemic stroke; Pre-MI, previous myocardial infarction; Pre-AF, previous atrial fibrillation; Pre-PCI, previous percutaneous coronary intervention; RDW, red blood cell distribution width; STEMI, ST-elevated myocardial infarction; TC, total cholesterol; TG, triglycerides; UA, uric acid; WBC, white blood cell.

**Table 3 nutrients-16-03176-t003:** The univariate and multivariate Cox regression analyses of all-cause mortality.

Variable	Univariate			Multivariate		
	HR	95% CI	*p*-Value	HR	95% CI	*p*-Value
Age, per 1 year	1.07	1.06, 1.09	<0.001	1.05	1.03, 1.06	<0.001
Age ≥ 65	2.86	2.11, 3.85	<0.001	2.13	1.76, 3.06	<0.001
Sex, male as reference	0.86	0.66, 1.13	0.300			
MI	3.68	2.82, 4.82	<0.001	2.20	1.66, 2.91	<0.001
Pre-MI	1.42	1.02, 1.99	0.039			
Pre-AF	2.20	1.37, 3.51	0.001			
Hypertension	1.67	1.20, 2.33	0.002			
DM	1.29	1.00, 1.68	0.052			
Dyslipidemia	0.86	0.56, 1.30	0.500			
Pre-IS	2.06	1.54, 2.76	<0.001	1.52	1.13, 2.04	0.005
PAD	1.69	0.75, 3.79	0.200			
CKD	5.75	4.17, 7.94	<0.001	1.83	1.28, 2.62	<0.001
Current smoker	1.03	0.77, 1.37	0.800			
RDW, RDW ≤ 13.6 as reference	3.49	2.69, 4.53	<0.001			
PNI, PNI > 44.2 as reference	6.92	5.34, 8.98	<0.001			
Hb, Hb ≤ 90 g/L as reference	0.11	0.08, 016	<0.001	0.42	0.28, 0.59	<0.001
Hb	0.96	0.96, 0.97	<0.001			
Alb	0.82	0.81, 0.84	<0.001	0.94	0.91, 0.97	<0.001
H-PNI/L-RDW	-					
L-PNI/L-RDW	7.73	4.88, 10.86	<0.001	3.96	2.60, 6.00	<0.001
H-PNI/H-RDW	3.68	2.45, 5.50	<0.001	3.00	1.99, 4.50	<0.001
L-PNI/H-RDW	18.50	12.83, 26.68	<0.001	8.85	5.96, 13.15	0.020

Abbreviations: Alb, serum albumin; CKD, chronic kidney disease; DM, diabetes; Hb, hemoglobin; MI, myocardial infarction; PNI, prognostic nutritional index; PAD, peripheral artery disease; Pre-IS, previous ischemic stroke; Pre-MI, previous myocardial infarction; Pre-AF, previous atrial fibrillation; RDW, red blood cell distribution width.

**Table 4 nutrients-16-03176-t004:** The association of the prognostic nutrition index and red blood cell distribution width with clinical outcomes.

Outcome	Events/Total	Crude HR (95% CI)	*p*	Adjusted HR (95% CI)	*p*
1-year ACM	235/5605				
H-PNI/L-RDW	43/3431	reference		reference	
H-PNI/H-RDW	52/1148	3.68 (2.45, 5.50)	<0.001	3.00 (1.99, 4.50)	<0.001
L-PNI/L-RDW	54/612	7.73 (4.88, 10.86)	<0.001	3.96 (2.60, 6.00)	<0.00
L-PNI/R-RDW	86/414	18.5 (12.83, 26.68)	<0.001	8.85 (5.96, 13.15)	0.02
MACCEs	560/5605				
H-PNI/L-RDW	219/3431	reference		reference	
H-PNI/H-RDW	144/1148	2.04 (1.65, 2.52)	<0.001	1.81 (1.46, 2.24)	<0.001
L-PNI/L-RDW	92/612	2.49 (1.96, 3.18)	<0.001	1.74 (1.35, 2.25)	<0.001
L-PNI/R-RDW	105/414	4.49 (3.56, 5.68)	<0.001	2.74 (2.12, 3.54)	<0.001

Abbreviations: ACM, all-cause mortality; CI, confidence interval; HR, adjusted hazards ratio; H, high; L, low; MACCEs, major adverse cardiac cerebrovascular events; PNI, prognostic nutritional index; RDW, red blood cell distribution width.

## Data Availability

The datasets used and/or analyzed in the current study are available from the corresponding author upon reasonable request.

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
