# Peer review of "The Combination Effect of the Red Blood Cell Distribution Width and Prognostic Nutrition Index on the Prognosis in Patients Undergoing PCI"

_nutrients, 2024, doi:10.3390/nu16183176_

Round 1

Reviewer 1 Report

Comments and Suggestions for Authors

The manuscript entitled The combination effect of RDW and PNI on the prognosis for CAD patients undergoing PCI is an original article, which is well written. However, it has some very important issues.

The patients included in the study were tested for SARSCOV2? This is a major cornerstone of this study. Between 2019 and 2021 was COVID-19 pandemia. These patients could have lymphopenia and other red blood cell changes. Moreover, hypoalbuminemia is associated with increased risk of thrombosis. If in these patients were not excluded a SARSCOV 2 infection, all data are compromised because the whole methodology is wrong.

Reviewer 2 Report

Comments and Suggestions for Authors

This study aimed to explore the clinical value of combination of RDW and PNI on admission, in predicting clinical outcomes after PCI, with respect to improve the risk stratification.

Inflammation and malnutrition are related to adverse clinical outcomes in patients with coronary artery disease (CAD). Nutritional status is an important condition that affects inflammation.

PNI was calculated as serum albumin (g / L) + 5 ×total lymphocyte count.

Red cell distribution width (RDW) is easy to obtain and represents the coefficient of variation of red blood cell volume distribution width. Many mechanisms can affect RDW levels, such as inflammatory stress, adrenergic activation, nutritional deficiency and iron homeostasis disorder. Previous studies showed that increased RDW is a powerful independent predictor of cardiovascular events in heart disease patients.

The coexistence of PNI and RDW appears to have a synergistic effect, providing further information for the risk stratification of CAD patients.

The article turns out to be well-written, the data exposed are also statistically significant.

The English vocabulary is adequate.

Limitations are highlighted and it would be appropriate to reevaluate these studies in view of the variation in PNI and RDW values.

Comments on the Quality of English Language

The English vocabulary is adequate.

Reviewer 3 Report

Comments and Suggestions for Authors

The authors present an important topic of the combination effect of RDW and PNI on the prognosis for CAD patients undergoing PCI. This study demonstrated that the combination of PNI and RDW was a strong predictor of 1- year ACM. Furthermore, the  CAD patients with L-PNI and H-RDW experienced the worst prognosis. The coexistence of PNI and RDW appears to have a synergistic effect, providing further information for the risk stratification of CAD patients. This study has clinical implications but the clinical usefulness of the above findings requires confirmation in prospective and multicenter studies with a large number of patients. The study has been well planned, the methodology and statistics are adequate. However, I have a following comments. 

Specific comments in detail:

1.     Abbreviations such as PNI and RDW should be expanded on first use

2.     The introduction is too superficial-please include the exact mechanisms.

3.     Please complete the units for the laboratory parameters in Table 1. The hemoglobin value seems to be incorrect.

4.     In Table 2, please expand the abbreviations of the table titles. The table is not legible

5.     Table 4 is illegible, please complete the abbreviations.

Round 2

Reviewer 1 Report

Comments and Suggestions for Authors

Thank you for responding to my comments.

Reviewer 3 Report

Comments and Suggestions for Authors

Thank you for your answers. I accept the manuscript in this current form.